# Pix2Key: Controllable Open-Vocabulary Retrieval with Semantic Decomposition and Self-Supervised Visual Dictionary Learning

**Guoyizhe Wei** [1]  **Yang Jiao** [2]  **Nan Xi** [2]  **Zhishen Huang** [2]  **Jingjing Meng** [2]  **Rama Chellappa** [1]  **Yan Gao** [2]

## Abstract

Composed Image Retrieval (CIR) uses a reference image plus a natural-language edit to retrieve images that apply the requested change while preserving other relevant visual content. Classic fusion pipelines typically rely on supervised triplets and can lose fine-grained cues, while recent zero-shot approaches often caption the reference image and merge the caption with the edit, which may miss implicit user intent and return repetitive results. We present Pix2Key, which represents both queries and candidates as open-vocabulary visual dictionaries, enabling intent-aware constraint matching and diversity-aware reranking in a unified embedding space. A self-supervised pretraining component, V-Dict-AE, further improves the dictionary representation using only images, strengthening fine-grained attribute understanding without CIR-specific supervision. On the DFMM-Compose benchmark, Pix2Key improves Recall@10 up to 3.2 points, and adding V-Dict-AE yields an additional 2.3-point gain while improving intent consistency and maintaining high list diversity.

## 1. Introduction

Composed image retrieval (CIR) is a multimodal search problem where a query consists of a reference image and a natural-language edit, and the system retrieves images that realize the requested change while preserving other relevant visual content. This interaction mirrors how users search in practice: shoppers seek the same garment with a different fabric or pattern, creators look for scene variations under different conditions, and designers search for layouts with localized modifications. Compared to standard text-to-image retrieval, CIR is conditional and fine-grained: it requires understanding what is meant to change, what should remain invariant, and which details define identity. Classical CIR systems are commonly trained with triplets built from reference, edit, and target images, and learn an explicit composition function for combining visual and textual signals (Vo et al., 2019). While such supervision can be effective, it is expensive to scale and can encourage a single fused representation that implicitly decides which fine-grained attributes to preserve, often in a non-transparent manner.

Recent progress in large-scale vision–language pretraining has enabled alternatives that reduce reliance on CIR-specific supervision. Contrastive pretraining in particular yields aligned image and text embeddings that generalize across domains (Radford et al., 2021). This has motivated a plethora of zero-shot CIR methods that repurpose pretrained models without triplet training. A representative line maps the reference image into a learnable textual token and composes it with the edit to form a query (Baldrati et al., 2023). Another popular practice uses a vision–language model as an image captioner, rewrites the caption according to the edit, and retrieves by matching in the language space. Despite their practicality, many zero-shot pipelines still struggle with subtle edits and localized attributes. Collapsing an image into a single token or a single sentence creates a lossy bottleneck: missing a small detail such as neckline shape, sleeve type, or a local pattern can invalidate an otherwise plausible result. In addition, ranking by similarity to a single fused query embedding often yields homogeneous top results, where near-duplicates crowd out diverse yet valid candidates. Diversity-aware reranking has a long history in information retrieval (Carbonell & Goldstein, 1998), but is rarely coupled with an intent representation that makes constraint satisfaction controllable in zero-shot CIR.

A persistent limitation lies not only in modeling, but also in evaluation. Attribute-centric datasets typically offer structured labels but lack natural-language edits, making them unsuitable for testing language-driven intent. In contrast, popular CIR benchmarks provide reference–target pairs and edit text, yet do not include fine-grained attributes for measuring how well the returned list satisfies the requested constraints beyond the single labeled target. This mismatch

---

[1]Johns Hopkins University, Baltimore, US. [2]Amazon.com, Seattle, US. . Correspondence to: Yan Gao <yan-ngao@amazon.com>.

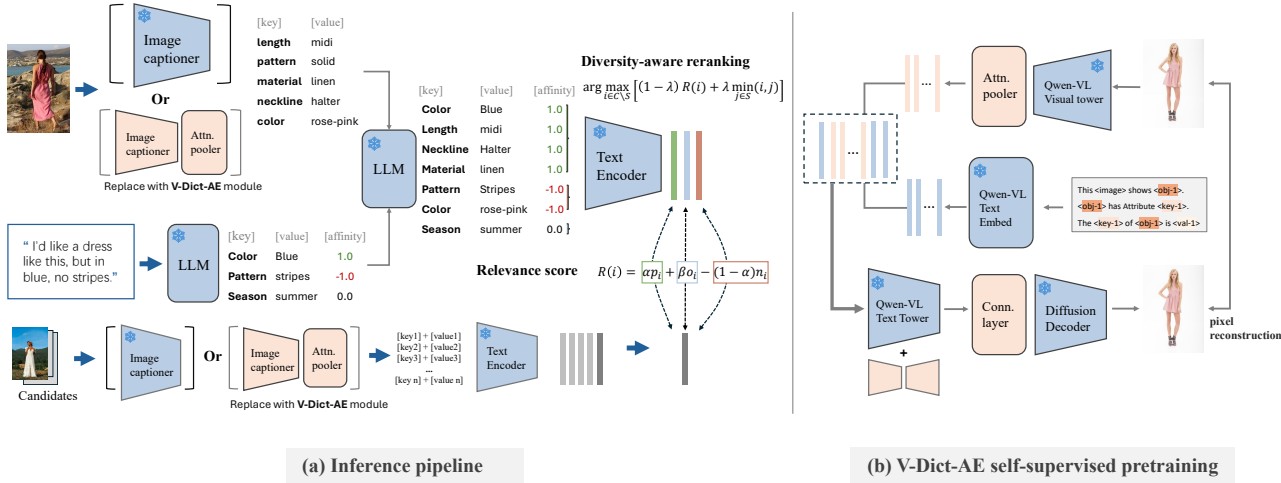

*Figure 1.* Overview of Pix2Key. (a) Inference pipeline: both the composed query and candidate images are converted into visual dictionaries for unified matching, followed by diversity-aware reranking. (b) V-Dict-AE pretraining: a self-supervised autoencoding objective learns compact visual-dictionary tokens by reconstructing images through a frozen generative decoder, improving fine-grained intent alignment for retrieval. The pretrained VLM can replace the captioner in the inference pipeline for dictionary extraction.

prevents quantifying how many non-target candidates in the top-ranked list actually satisfy the user's requirements, and makes it difficult to analyze similarity and diversity in an attribute-grounded way. The problem is further compounded by annotation noise: in many CIR datasets, the designated target is not necessarily the best match to the edit, which can cause supervised methods to learn spurious correlations or be penalized for retrieving a better solution than the labeled target.

Pix2Key is introduced to address these challenges with a two-part design that remains free of CIR-specific triplet supervision while improving fine-grained controllability. First, images are represented as compact visual dictionaries, and edits are decomposed into structured constraints that explicitly separate attributes to satisfy, attributes to avoid, and attributes left underspecified by the user intent. The same dictionary representation is applied to the candidate database, turning retrieval into matching between two structured descriptions rather than fragile cross-modal fusion. A lightweight diversity-aware reranking stage then exposes a user-facing trade-off between strict constraint satisfaction and result variety, enabling multiple plausible completions when an edit admits more than one valid outcome. Second, V-Dict-AE improves the faithfulness of dictionary representations via self-supervised pretraining: a visual-dictionary autoencoder is trained to encode images into compact token sequences aligned with a frozen text encoder and a frozen diffusion decoder (Rombach et al., 2022). This training uses only images, and adapts a limited set of parameters through efficient low-rank updates (Hu et al., 2022), encouraging the representation to preserve the visual evidence most relevant to fine-grained retrieval.

To enable attribute-grounded evaluation of both intent satisfaction and list diversity, an additional contribution is a derived benchmark DFMM-Compose built on DeepFashion-MM (Jiang et al., 2022). It augments structured attribute labels with generated edit descriptions and auxiliary tags so that CIR queries can be evaluated not only by whether a single target is retrieved, but also by how consistently the top-ranked list satisfies the intended attributes and how diverse the returned candidates are. Together, these components support a CIR system that is more controllable, more interpretable, and more measurable under realistic, noisy supervision.

Our contributions are:

- Pix2Key, a training-free CIR framework that represents queries and candidates as visual dictionaries, making fine-grained intent constraints explicit and controllable.

- A diversity-aware reranking mechanism integrated with the dictionary-based intent representation, enabling trade-offs between constraint satisfaction and result diversity.

- V-Dict-AE, a self-supervised visual-dictionary autoencoder aligned with frozen text and diffusion components, preserving fine-grained evidence without CIR triplets.

- An attribute-grounded CIR benchmark DFMM-Compose that supports quantitative evaluation of intent satisfaction and list diversity under natural-language edits.

## 2. Related Work

**Composed image retrieval.** CIR studies retrieval where a query combines a reference image and an edit instruction. Early supervised methods learn an explicit composition function under triplet objectives (Vo et al., 2019). Later work improves image–text fusion and matching for text-feedback retrieval, including visiolinguistic attention and explicit matching (Chen et al., 2020; Delmas et al., 2022), joint visual–semantic embeddings (Chen & Bazzani, 2020), compositional query learning (Anwaar et al., 2021), and CLIP-feature-based conditioned composition (Baldrati et al., 2022). Related advances also refine the shared retrieval space with normalization or metric-learning formulations (Bogolin et al., 2022; Roth et al., 2022a;b) and explore composed retrieval in zero-shot protocols (Liu et al., 2023b), while content–style modulation further improves modeling of preserved vs. edited factors (Lee et al., 2021). Benchmarks such as FashionIQ and CIRR standardize evaluation with reference–target pairs (Wu et al., 2021; Liu et al., 2021), building on broader fine-grained fashion representation learning (Jiao et al., 2022; 2023), but supervision is often tied to a single labeled target, limiting intent assessment beyond target hit.

**Tokenization-based zero-shot CIR.** Zero-shot CIR often reuses a frozen CLIP-style retriever and represents the reference image as learnable tokens inside the text encoder. Pic2Word maps the image to a pseudo-word appended to the edit, enabling retrieval without CIR-specific training (Saito et al., 2023; Radford et al., 2021). SEARLE and iSEARLE cast this mapping as textual inversion, optimizing pseudo-tokens so the composed embedding aligns with the target (Baldrati et al., 2023; Gal et al., 2023; Agnolucci et al., 2025), and related token-personalization strategies are explored in PALAVRA (Cohen et al., 2022). To reduce per-query optimization or increase expressiveness, FTI4CIR amortizes inversion, Context-I2W conditions tokenization on the edit context, and ISA/LinCIR move from single tokens toward sentence-level prompts while keeping CLIP-compatible indexing (Zhang et al., 2024; Tang et al., 2024; Du et al., 2024; Gu et al., 2024); prompt-centric and latent textual prompt variants further support composed retrieval and domain adaptation in vision–language models (Bai et al., 2024; Wang et al., 2024a; Wei et al., 2025). A remaining limitation is that multiple constraints must be compressed into a small token budget, while ranking still relies on a single fused similarity score.

**Training-free inference with large VLMs.** Another training-free line translates visual evidence into language at inference time and retrieves in text space. CIReVL captions the reference image with a VLM, rewrites the caption conditioned on the edit, and matches the rewritten text to

candidates (Karthik et al., 2024). This avoids per-query optimization and yields an interpretable intermediate query, but performance depends on caption coverage and rewriting stability, so omitted attributes or underspecified invariants can cause drift. Such pipelines build on foundation VLMs and image–text pretraining, including CLIP-style encoders and instruction-tuned multimodal models (Radford et al., 2021; Li et al., 2023; Liu et al., 2023a; Bai et al., 2025; Wang et al., 2024c; Yu et al., 2022; Singh et al., 2022). They are also related to recent advances in visual and vision–language representation learning, including efficient visual modeling, domain-adaptive prompting, and dense vision–language inference (Wei & Chellappa, 2025; Wang et al., 2024b; 2026).

## 3. Method

### 3.1. Problem Setup

Pix2Key targets composed image retrieval (CIR), where a query is formed by a reference image together with a natural-language edit. The goal is to retrieve images that satisfy the edit while preserving other relevant visual content. Pix2Key uses an open-vocabulary visual dictionary as a shared interface for both queries and database images, so retrieval can be implemented as similarity search in a text embedding space. A self-supervised visual-dictionary autoencoder, V-Dict-AE, further refines the dictionary representation without requiring CIR triplets.

Let the retrieval database be $\mathcal{I} = \{I_i\}_{i=1}^N$. A composed query is a pair $(I_q, T)$, where $I_q$ is the reference image and $T$ is a free-form edit instruction. The system returns a ranked list $\pi$ over $\mathcal{I}$. Throughout the paper, cosine distance is used for nearest-neighbor search,

$$\text{dist}(\mathbf{x}, \mathbf{y}) = 1 - \text{cossim}(\mathbf{x}, \mathbf{y}). \quad (1)$$

This distance is equivalent to cosine similarity up to an order-preserving transformation, and matches the implementation used in retrieval and reranking.

### 3.2. Open-Vocabulary Visual Dictionaries

Each gallery image is converted into an open-vocabulary dictionary of attribute-like facts,

$$\mathcal{D}_{\text{img}}(I) = \{(k_m, v_m)\}_{m=1}^M. \quad (2)$$

where $k_m$ is an attribute key (e.g., `color`, `pattern`) and $v_m$ is its value (e.g., `red`, `striped`). A composed query is represented by a signed dictionary

$$\mathcal{D}_q = \{(k_m, v_m, p_m)\}_{m=1}^{M_q}, \qquad p_m \in \{+1, 0, -1\}. \quad (3)$$

The intent polarity $p_m$ is defined only on the query side: $p_m = +1$ indicates desired attributes (add/strengthen),

| Method | Dresses | | Shirts | | Tops&Tees | | Avg | |
|---|---|---|---|---|---|---|---|---|
| | R@10 | R@50 | R@10 | R@50 | R@10 | R@50 | R@10 | R@50 |
| *Training-free methods:* | | | | | | | | |
| Image-only | 4.76 | 12.35 | 7.07 | 15.82 | 6.87 | 14.64 | 6.23 | 14.27 |
| Text-only | 14.86 | 34.14 | 19.57 | 33.79 | 21.17 | 39.39 | 18.53 | 35.77 |
| Image + Text | 13.57 | 31.27 | 14.96 | 26.70 | 19.35 | 33.74 | 15.96 | 30.57 |
| CIReVL (Karthik et al., 2024) | 23.74 | 45.62 | 29.01 | 47.65 | 30.87 | 52.76 | 27.87 | 48.68 |
| Pix2Key | 24.92 | 47.19 | 30.62 | 49.64 | 32.00 | 54.61 | 29.18 | 50.48 |
| *With pretrained tokenization:* | | | | | | | | |
| Pic2Word (Saito et al., 2023) | 20.00 | 40.20 | 26.20 | 43.60 | 27.90 | 47.40 | 24.70 | 43.70 |
| PALAVRA (Cohen et al., 2022) | 17.25 | 35.94 | 21.49 | 37.05 | 20.55 | 38.76 | 19.76 | 37.25 |
| SEARLE (Baldrati et al., 2023) | 20.48 | 43.13 | 26.89 | 45.58 | 29.32 | 49.97 | 25.56 | 46.23 |
| FTI4CIR (Zhang et al., 2024) | 24.39 | 47.84 | 31.35 | 50.59 | 32.43 | 54.21 | 29.39 | 50.88 |
| Pix2Key+V-Dict-AE | **25.61** | **48.92** | **31.69** | **51.43** | **32.58** | **55.39** | **29.96** | **51.91** |

*Table 1.* **FashionIQ** composed image retrieval results, reported as Recall@10 and Recall@50 on Dresses, Shirts, and Tops&Tees, as well as the overall average. *Image-only*, *Text-only*, and *Image+Text* denote unimodal and naive fusion baselines under the standard evaluation protocol. Pix2Key variants are highlighted in orange. Best in each column is in **bold**.

$p_m = -1$ indicates attributes to avoid (remove/contradict), and $p_m = 0$ denotes open-set anchors that are not explicitly constrained but help preserve salient context from the reference image.

A vision-language model extracts $\mathcal{D}_{\text{img}}(I)$ from an image using a constrained prompt format that encourages short key–value descriptions. In experiments, Qwen-VL style models are used as the extractor due to strong visual grounding and attribute coverage (Bai et al., 2025; Wang et al., 2024c). For a composed query $(I_q, T)$, we first extract $\mathcal{D}_{\text{ref}} = \mathcal{D}_{\text{img}}(I_q)$, then decompose the edit text into signed updates $\Delta\mathcal{D}(T) = \{(k, v, p)\}$, and finally merge them:

$$\mathcal{D}_q = \text{Merge}(\mathcal{D}_{\text{ref}}, \Delta\mathcal{D}(T)), \tag{4}$$

where edits override conflicting reference entries on the same key, negative entries are kept as explicit constraints, and unconstrained entries can serve as anchors that encourage preservation when the edit underspecifies invariants.

### 3.3. Text-Space Indexing from Dictionaries

Pix2Key converts database images into dictionary text offline and indexes them in a text embedding space. Compared to caption-based pipelines, we serialize only attribute-like key–value facts to reduce nuisance details and expose a controllable interface via intent polarity, while still enabling efficient offline indexing with a single text embedding per gallery item.

A dictionary is serialized into a short string $\text{ser}(\mathcal{D})$ (e.g., `key:value; key:value; ...`). A frozen text encoder $f_{\text{text}}$ maps the serialized dictionary into a global embedding:

$$\mathbf{e}_i = f_{\text{text}}\big(\text{ser}(\mathcal{D}(I_i))\big) \in \mathbb{R}^d. \tag{5}$$

In practice, $f_{\text{text}}$ is an OpenCLIP text encoder initialized

from public pretrained weights; all $\mathbf{e}_i$ are precomputed and stored for fast nearest-neighbor search.

For queries, $\mathcal{D}_q$ is split by polarity and each subset is serialized and embedded:

$$\text{ser}(\mathcal{D}_q^+), \qquad \text{ser}(\mathcal{D}_q^0), \qquad \text{ser}(\mathcal{D}_q^-). \tag{6}$$

$$\mathbf{q}^+, \ \mathbf{q}^0, \ \mathbf{q}^- \ \in \ \mathbb{R}^d. \tag{7}$$

This keeps the retrieval space unified (queries and candidates share the same text embedding space) while preserving polarity-aware control.

### 3.4. Intent-Aware Relevance Scoring

Given a candidate embedding $\mathbf{e}_i$ and query embeddings from Eq. equation 7, Pix2Key computes aligned similarity terms:

$$\begin{aligned} p_i &= \text{cossim}(\mathbf{q}^+, \mathbf{e}_i), \\ o_i &= \text{cossim}(\mathbf{q}^0, \mathbf{e}_i), \\ n_i &= \text{cossim}(\mathbf{q}^-, \mathbf{e}_i). \end{aligned} \tag{8}$$

A single scalar relevance score is formed as

$$R(i) = \alpha\, p_i + \beta\, o_i - (1-\alpha)\, n_i, \tag{9}$$

where $\alpha$ balances enforcing requested changes against suppressing forbidden attributes, and $\beta$ controls how strongly unconstrained anchors are preserved.

### 3.5. Diversity-Aware Reranking

Relevance-only ranking often returns near-duplicates, so Pix2Key applies a diversity-aware reranker over a candidate pool $\mathcal{C}$ (see Appendix C for the full procedure). Let $\mathbf{e}_i$ denote the global embedding of candidate $i$, and define pairwise cosine distance as in Eq. equation 1:

$$\text{dist}(i, j) = 1 - \text{cossim}(\mathbf{e}_i, \mathbf{e}_j). \tag{10}$$

| Method | CIRR | | | | DFMM-Compose | | | |
|---|---|---|---|---|---|---|---|---|
| | R@1 | R@5 | R@10 | R@50 | AC@50 | ILD@50 | R@10 | R@50 |
| *Training-free methods:* | | | | | | | | |
| CIReVL (Karthik et al., 2024) | 23.94 | 52.51 | 66.00 | 86.95 | 36.42 | 46.44 | 18.12 | 35.40 |
| CIReVL+MMR | 25.17 | 52.93 | 66.12 | 86.45 | 35.79 | 49.26 | 19.30 | 34.82 |
| Pix2Key | 27.02 | 54.26 | 68.15 | 89.44 | 51.26 | **54.15** | 21.31 | 37.56 |
| *With pretrained tokenization:* | | | | | | | | |
| Pic2Word (Saito et al., 2023) | 23.90 | 51.72 | 65.30 | 87.82 | 33.56 | 46.82 | 16.89 | 32.06 |
| SEARLE (Baldrati et al., 2023) | 24.20 | 52.40 | 66.30 | 88.60 | 35.75 | 46.19 | 18.94 | 37.35 |
| FTI4CIR (Zhang et al., 2024) | 25.90 | 55.61 | 67.66 | 89.66 | 38.17 | 47.24 | 19.35 | 37.22 |
| Context-I2W (Tang et al., 2024) | 25.60 | 55.10 | 68.50 | 89.80 | 39.59 | 45.56 | 20.87 | 37.15 |
| Pix2Key+V-Dict-AE | **29.06** | **59.44** | **73.36** | **92.08** | **54.44** | 53.96 | **23.58** | **40.96** |

*Table 2.* Results on **CIRR** and **DFMM-Compose**. CIRR is evaluated by Recall@K. DFMM-Compose reports Recall@K together with two list-level metrics computed over the top-50 retrieved candidates: AC@50 for attribute consistency and ILD@50 for intra-list diversity. Pix2Key variants are highlighted in orange. Best in each column is in **bold**.

A greedy selection set $S$ is built by repeatedly choosing

$$i^\star \;=\; \arg\max_{i \in \mathcal{C} \setminus S} \left[ (1-\lambda)\, R(i) \;+\; \lambda \min_{j \in S} \operatorname{dist}(i,j) \right], \quad (11)$$

where $\lambda$ is a user-facing diversity control. Before reranking, we linearly normalize $R(i)$ to $[0,1]$ over $\mathcal{C}$, and rescale cosine distance as $\bar{d}(i,j) = \operatorname{dist}(i,j)/2$ for a consistent tradeoff across queries. This distance-form is equivalent to the classic similarity-form MMR objective up to a constant shift (Carbonell & Goldstein, 1998).

### 3.6. V-Dict-AE: Self-Supervised Visual Dictionary Autoencoder

Dictionary extraction and query parsing can miss fine-grained cues. V-Dict-AE improves dictionary token quality using only unlabeled images by training a parameter-efficient image-to-slot module supervised through a frozen diffusion decoder, encouraging the token sequence to preserve visually salient details needed for reconstruction. The diffusion model, VAE, and the text encoder used by diffusion are frozen; only lightweight modules are trained.

Given an image $I$, a frozen visual tower extracts patch-level features:

$$\mathbf{P} \;=\; g_{\text{vis}}(I) \in \mathbb{R}^{B \times N \times h}, \quad (12)$$

where $B$ is batch size, $N$ is the number of visual tokens, and $h$ is the VLM hidden dimension. We obtain a fixed-length slot sequence using an attention pooler with $Q$ learnable queries $\mathbf{Q}_0 \in \mathbb{R}^{Q \times h}$ replicated across the batch. The pooler concatenates queries and patch tokens and applies a Transformer encoder:

$$\mathbf{X} \;=\; \operatorname{Enc}\big([\mathbf{Q}_0; \mathbf{P}]\big) \in \mathbb{R}^{B \times (Q+N) \times h}. \quad (13)$$

Let $\mathbf{X} = [\mathbf{x}_1, \dots, \mathbf{x}_{Q+N}]$; we take the first $Q$ tokens as pooled slots:

$$\mathbf{Z} \;=\; [\mathbf{x}_1, \dots, \mathbf{x}_Q] \in \mathbb{R}^{B \times Q \times h}. \quad (14)$$

To align slot features with language semantics, slots are injected into the frozen VLM in place of its image placeholder tokens. Let $\mathbf{y}$ be the token ids of a prompt containing one image placeholder; the placeholder position is expanded to $Q$ repeated image-token ids to form $\tilde{\mathbf{y}}$. With the VLM token embedding layer $\operatorname{Emb}(\cdot)$, we replace the embeddings at image-token positions by pooled slots:

$$\mathbf{E} \;=\; \operatorname{Emb}(\tilde{\mathbf{y}}), \qquad \mathbf{E}_{\text{img}} \leftarrow \mathbf{Z}. \quad (15)$$

A forward pass through the frozen VLM yields hidden states $\mathbf{H} \in \mathbb{R}^{B \times L \times h}$, from which we obtain $Q$ slot-conditioned vectors $\mathbf{H}_{\text{dict}} \in \mathbb{R}^{B \times Q \times h}$ (by taking image-token positions, or by short greedy decoding and padding/truncation).

Latent diffusion models are commonly conditioned by a CLIP-like text transformer (Radford et al., 2021; Rombach et al., 2022); V-Dict-AE maps each slot into the CLIP token embedding space. Let $d$ be the CLIP token embedding dimension. A continuous projection head produces

$$\mathbf{u}_m \;=\; \operatorname{LN}(\mathbf{W}\,\mathbf{h}_m) \quad \text{for } m = 1, \dots, Q, \quad (16)$$

where $\mathbf{h}_m$ is the $m$-th slot vector in $\mathbf{H}_{\text{dict}}$, $\mathbf{W} \in \mathbb{R}^{d \times h}$ is trainable, and LN is LayerNorm. A vocabulary-distributed variant predicts CLIP vocabulary logits and takes an expected embedding. Let $|\mathcal{V}|$ be the CLIP vocabulary size and $\mathbf{E}_{\text{CLIP}} \in \mathbb{R}^{|\mathcal{V}| \times d}$ be the frozen CLIP token embedding matrix:

$$\begin{aligned} \boldsymbol{\ell}_m &= \mathbf{W}_{\mathcal{V}}\,\mathbf{h}_m, \\ \mathbf{p}_m &= \operatorname{softmax}(\boldsymbol{\ell}_m/\tau), \quad (17) \\ \mathbf{u}_m &= \mathbf{p}_m^\top \mathbf{E}_{\text{CLIP}}. \end{aligned}$$

A temperature schedule sharpens the distribution during training:

$$\tau \leftarrow \max(\tau_{\min}, \eta\,\tau), \quad (18)$$

with decay $\eta \in (0,1)$. A length-$Q+2$ soft prompt is formed by adding frozen beginning and end token embeddings:

$$\mathbf{U} \;=\; [\mathbf{u}_{\text{BOS}}; \mathbf{u}_1; \dots; \mathbf{u}_Q; \mathbf{u}_{\text{EOS}}] \in \mathbb{R}^{B \times (Q+2) \times d}. \quad (19)$$

A frozen latent diffusion model supplies the self-supervised signal. The input image $I$ is encoded by a frozen VAE into a latent $\mathbf{x}_0$. A diffusion timestep $t$ and Gaussian noise $\boldsymbol{\epsilon}$ are sampled, producing

$$\mathbf{x}_t \;=\; \sqrt{\alpha_t}\,\mathbf{x}_0 \;+\; \sqrt{1-\alpha_t}\,\boldsymbol{\epsilon}, \qquad (20)$$

where $\alpha_t$ denotes the cumulative noise schedule. The diffusion model is conditioned on a context produced from $\mathbf{U}$:

$$\mathbf{c} \;=\; \text{CLIPText}(\mathbf{U}). \qquad (21)$$

Following the v-parameterization (Salimans & Ho, 2022), the target is

$$\mathbf{v}_t \;=\; \sqrt{\alpha_t}\,\boldsymbol{\epsilon} \;-\; \sqrt{1-\alpha_t}\,\mathbf{x}_0. \qquad (22)$$

A frozen UNet predicts $\hat{\mathbf{v}}_\theta$ from $(\mathbf{x}_t, t, \mathbf{c})$:

$$\hat{\mathbf{v}}_\theta \;=\; \text{UNet}(\mathbf{x}_t, t, \mathbf{c}). \qquad (23)$$

The main training loss is

$$\mathcal{L}_v \;=\; \mathbb{E}_{t,\boldsymbol{\epsilon}}\Big[\|\hat{\mathbf{v}}_\theta - \mathbf{v}_t\|_2^2\Big]. \qquad (24)$$

From the v-parameterization, an estimate of the clean latent is

$$\hat{\mathbf{x}}_0 \;=\; \sqrt{\alpha_t}\,\mathbf{x}_t \;-\; \sqrt{1-\alpha_t}\,\hat{\mathbf{v}}_\theta. \qquad (25)$$

Decoding with the frozen VAE gives $\hat{I} = \text{VAE}^{-1}(\hat{\mathbf{x}}_0)$ and a lightweight pixel loss

$$\mathcal{L}_{\text{pix}} \;=\; \|\hat{I} - I\|_1. \qquad (26)$$

The final objective is

$$\mathcal{L} \;=\; \mathcal{L}_v \;+\; \gamma\,\mathcal{L}_{\text{pix}}. \qquad (27)$$

In practice, $\gamma$ is small so the diffusion loss remains the primary signal while the pixel loss stabilizes reconstruction.

Only parameter-efficient modules are trained relative to full finetuning: the attention pooler, the projection head, and optional low-rank adapters inserted into the frozen VLM (Hu et al., 2022). After training, the learned pooler and adapters are reused by the dictionary extractor at inference by replacing raw patch embeddings with pooled slots from Eq. equation 14, improving fine-grained attribute capture while preserving the retrieval interface in Sections 3.2–3.5.

## 4. Experiments

### 4.1. Experimental Setting

**Overview.** Pix2Key is an open-vocabulary dictionary retrieval system for CIR. At inference time, a pretrained vision–language model converts the reference image and gallery images into compact visual dictionaries, and the edit

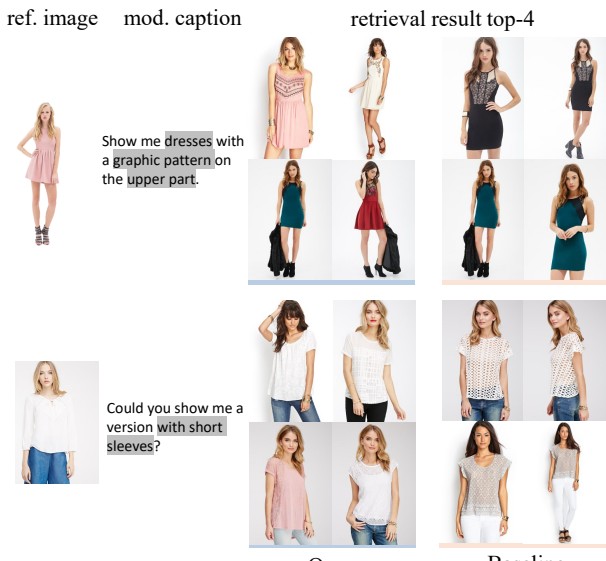

*Figure 2.* Qualitative comparison of composed retrieval results. Each example shows the reference image, the modification text, and the top-4 retrieved candidates.

text is decomposed into polarity-aware constraints. Both query and gallery dictionaries are embedded with an Open-CLIP text encoder pretrained on LAION-2B (Ilharco et al., 2021), enabling nearest-neighbor retrieval in a shared text space, followed by MMR reranking for controllable diversity. V-Dict-AE is an optional self-supervised pretraining module that improves the dictionary representation by training a parameter-efficient slot encoder on COCO2017 (Lin et al., 2014) or FashionAI (Zou et al., 2019), while keeping the same inference-time retrieval interface.

Evaluation covers FashionIQ (Wu et al., 2021) and CIRR (Liu et al., 2021) for standard Recall@K, and DFMM-Compose for attribute-grounded intent satisfaction and list diversity. Comparisons include tokenization-based zero-shot methods and training-free caption-rewrite pipelines. Full details of compute, model components and weights, pretraining configuration, benchmarks, baselines, and DFMM-Compose construction are provided in Appendix A.1.

**DFMM-Compose Benchmark.** DFMM-Compose is an attribute-grounded composed retrieval benchmark derived from DeepFashion-MM (Jiang et al., 2022). It is designed to evaluate not only whether the annotated target is retrieved, but also how well the returned candidate list satisfies fine-grained edit intent and how diverse the list remains. Each query consists of a reference image and a natural-language edit, while each gallery image is associated with structured attribute labels that enable intent-consistency scoring over the top-ranked results. This format supports standard Recall@K evaluation, attribute-consistency evaluation beyond a single target, and list-level diversity analysis for user-facing retrieval. Construction details and evaluation proto-

| Method | pos. | neg. | open | MMR | R@50 | ILD@50 | AC@50 |
|---|---|---|---|---|---|---|---|
| embeds pos. only | ✓ | ✗ | ✗ | ✗ | 37.27 | 51.65 | 52.49 |
| embeds pos. & neg. | ✓ | ✓ | ✗ | ✗ | 39.41 | 51.69 | 53.31 |
| embeds w/o neg. | ✓ | ✗ | ✓ | ✓ | 38.56 | 52.80 | 51.72 |
| w/o MMR reranking | ✓ | ✓ | ✓ | ✗ | 40.48 | 49.22 | 53.90 |
| Pix2Key+V-Dict-AE | ✓ | ✓ | ✓ | ✓ | **40.96** | **53.96** | **54.44** |

*Table 3*. Component ablations on **DFMM-Compose**. Columns *pos.*, *neg.*, and *open* indicate whether the query dictionary includes affirmative constraints, negated constraints, and open-set anchors, respectively; *MMR* indicates diversity-aware reranking. Our setting is highlighted in orange, and the best value in each metric column is in **bold**.

col are given in Appendix B.

**Metrics.** FashionIQ and CIRR are evaluated using Recall@K, which measures whether the annotated target appears among the top-K results. DFMM-Compose additionally reports two list-level metrics on the top-50 candidates: AC@50 quantifies how well returned images satisfy attribute changes implied by the edit, and ILD@50 measures redundancy within the list using attribute-based distances. Higher values indicate better performance across metrics. Full definitions are provided in Appendix D.

### 4.2. Main Results

**Accuracy.** Tables 1 and 2 summarize retrieval accuracy across FashionIQ, CIRR, and DFMM-Compose. On FashionIQ (Table 1), Pix2Key consistently improves over unimodal and naive fusion baselines, and is competitive among training-free approaches, outperforming CIReVL under the same Qwen2.5-VL backbone used for fair comparison (Appendix A.1). Notably, CIReVL represents the composed query by first generating an image caption for the reference and then rewriting it with the edit prompt, so the retrieval signal is mediated by a single free-form sentence in the text space. The consistent gap to Pix2Key therefore suggests that replacing caption-level descriptions with a structured key–value dictionary interface can yield a more stable and controllable representation for composed retrieval, especially when the edit requires fine-grained attribute changes.

Compared to methods that rely on pretrained tokenization or inversion-style modules, Pix2Key+V-Dict-AE achieves the strongest results across all FashionIQ categories and the overall average, indicating that self-supervised token refinement complements dictionary-based retrieval. On CIRR (Table 2), Pix2Key improves Recall@K over the training-free caption-rewrite baseline, and the pretrained variant further raises performance, yielding the best Recall@1/5/10/50 among all compared methods. On DFMM-Compose, the same trend holds: Pix2Key improves Recall@10/50 over prior baselines, and V-Dict-AE brings additional gains, suggesting that the representation improvement transfers beyond a single benchmark style and remains compatible with the same nearest-neighbor retrieval interface.

**Intent alignment.** DFMM-Compose enables attribute-grounded evaluation beyond target hit, since it provides fine-grained attribute labels for all gallery candidates, allowing us to quantify whether other retrieved items also satisfy the intended edit. As shown in Table 2, prior baselines yield relatively limited attribute consistency, while Pix2Key achieves substantially higher AC@50, suggesting that polarity-aware constraints and dictionary matching better capture fine-grained intent than caption-based rewriting or token-only fusion.

This difference is consistent with the fact that caption-rewrite pipelines (e.g., CIReVL) compress the reference evidence and the edit into a single rewritten sentence, which may under-specify attributes that should be preserved or suppressed, whereas the dictionary representation keeps explicit key–value evidence and separates desired, avoided, and open anchors. Adding V-Dict-AE further improves AC@50 together with Recall, indicating that reconstruction-shaped pretraining helps preserve the visual evidence that matters for downstream attribute-level satisfaction under composed edits.

**Diversity.** DFMM-Compose also supports list-level analysis because all candidates carry fine-grained attributes (and auxiliary tags), making it possible to measure redundancy and within-list variation in an interpretable space. In Table 2, Pix2Key achieves the highest ILD@50 among compared methods, indicating a less redundant candidate list under the same retrieval budget. Pix2Key+V-Dict-AE maintains similarly high ILD@50 while improving Recall@K and AC@50, suggesting that representation refinement does not collapse retrieval neighborhoods and remains compatible with diversity-aware reranking. This behavior is consistent with Pix2Key's design: intent is scored with polarity-aware relevance, while diversity is controlled at the list level via MMR, allowing multiple plausible results without drifting away from the edit intent.

**Additional robustness and efficiency.** Beyond the main benchmarks, we further examine Pix2Key under vague edit instructions, unseen categories, domain-shifted targets, smaller VLM backbones, and lexical variation in dictionary serialization. These analyses, reported in Appendix D.1, show that Pix2Key remains stable under noisy user inputs

| *(a)* Learning rate. | | *(b)* Pooler depth. | | *(c)* Input size. | |
|---|---|---|---|---|---|
| learn. rate | R@5 | depth | R@5 | input size | R@5 |
| $5 \times 10^{-5}$ | **59.44** | 3 | 59.01 | 64 | 58.01 |
| $1 \times 10^{-5}$ | 58.50 | 5 | **59.44** | 128 | 58.63 |
| $5 \times 10^{-6}$ | 57.93 | 9 | 56.90 | 224 | **59.44** |

| *(d)* Refined prompt. | | *(e)* LoRA. | | *(f)* Text encoder. | |
|---|---|---|---|---|---|
| setting | R@5 | setting | R@5 | setting | R@5 |
| original | 58.27 | w/o LoRA | 57.20 | ViT-B/32 | **59.44** |
| refined prompt | **59.44** | with LoRA | **59.44** | ViT-L/14 | 59.12 |

*Table 4.* Sensitivity analysis on **CIRR**, reported as Recall@5. Each subtable varies a single factor while keeping the remaining settings fixed. The default configuration in each sub-study is highlighted in orange, and the best value is shown in **bold**. Text encoder denotes the backbone used for embedding dictionary text into the retrieval space.

and moderate representation changes, while keeping query-time overhead practical for deployment.

### 4.3. Ablations

**Components.** Table 3 indicates that relying on affirmative constraints alone offers limited leverage for fine-grained edits, and yields the weakest Recall@50 together with a comparatively low AC@50. Adding explicit negation increases both measures, suggesting that describing attributes to avoid helps separate visually plausible but intent-violating candidates from true matches, especially when edits involve subtle attribute swaps. Introducing open-set anchors without negation tends to raise ILD@50, reflecting a broader and less redundant candidate set, but AC@50 drops in this setting, implying that anchors emphasize preserving general context and may weaken attribute specificity when conflicting cues are not suppressed. When affirmative, negative, and open-set signals are used together, Recall@50 increases further and AC@50 remains competitive, supporting the view that anchors are most effective when paired with explicit suppression to maintain a clearer notion of the intended change.

**Diversity control.** Table 3 also isolates the effect of diversity-aware reranking under the same intent representation. With MMR enabled, ILD@50 increases markedly, while Recall@50 and AC@50 remain stable and slightly improve in this ablation, suggesting that reranking can diversify the top-50 list without noticeably compromising intent satisfaction in this setting. Notably, applying MMR reranking to CIReVL slightly lowers Recall@50 on both CIRR and DFMM-Compose, suggesting that caption-and-rewrite representations provide a less stable relevance signal under diversity control, whereas Pix2Key's dictionary-based representation is more compatible with MMR and can diversify results without the same degree of ranking degradation.

**Sensitivity.** Table 4 suggests that Pix2Key is not overly brittle on CIRR under reasonable hyperparameter changes, as measured by Recall at five. A moderate learning rate performs best, indicating that the trainable dictionary modules benefit from sufficiently strong updates under a frozen backbone. Pooler depth shows a clear sweet spot in the middle range. Depth 5 achieves 59.44, depth 3 remains close at 59.01, and depth 9 drops to 56.90, which is consistent with the pooler acting as a lightweight summarizer rather than a full feature re encoder. Higher input resolution consistently improves performance. Increasing the resolution from 64 to 224 raises Recall at five from 58.01 to 59.44, suggesting that finer local evidence helps preserve subtle attributes under natural language edits. Prompt refinement also improves results, increasing the score from 58.27 to 59.44, and enabling LoRA provides an additional gain from 57.20 to 59.44, supporting the view that better extraction and targeted adaptation improve alignment without full finetuning. Finally, the OpenCLIP text encoder choice is stable in this study. ViT-B/32 reaches 59.44 and ViT-L/14 reaches 59.12. Overall, the strongest gains come from factors that improve fine grained evidence and representation alignment, which matches the intended design of Pix2Key and V-Dict-AE.

## 5. Conclusion

We introduce Pix2Key, a controllable composed image retrieval framework that represents both queries and candidates as open vocabulary visual dictionaries. The edit instruction is decomposed into intent signals that specify what to add, what to avoid, and what to keep open as anchors. This design provides an explicit and interpretable control interface, while reducing retrieval to fast nearest neighbor search in a shared text embedding space. To support user facing exploration, Pix2Key applies diversity aware reranking that increases variety among the top results without sacrificing relevance. We further propose V-Dict-AE, a parameter efficient self supervised module that refines dictionary slots through reconstruction based supervision, strengthening fine grained visual evidence without requiring

composed retrieval triplets. Experiments show consistent improvements in Recall at K, attribute consistency, and intra list diversity over strong zero shot baselines and competitive caption based retrieval pipelines. Pix2Key also enables attribute retrieval and diagnostic analysis through its dictionary interface. Overall, Pix2Key provides a practical and scalable approach to intent aware composed image retrieval.

## Impact Statement

This paper aims to advance machine learning research in composed image retrieval by improving alignment between a reference image and a natural-language edit, as well as the diversity and consistency of retrieved results. The expected positive impact is to enable more controllable and efficient search for benign applications such as e-commerce, creative design, and visual content organization.

Our method is trained with self-supervision on public datasets, which reduces reliance on human-provided labels and may lessen risks from annotation errors. However, the learned representations can still reflect biases in the underlying data distributions. Additional risks include retrieval of sensitive or copyrighted content, or privacy-invasive use when combined with external data sources. We encourage responsible deployment with content filtering, access control, dataset governance, and auditing practices. Overall, we do not anticipate ethical concerns beyond those commonly associated with vision-language retrieval systems, but responsible use remains important.

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

# A. Appendix.

## A.1. Additional Experimental Details

**Compute.** All reported retrieval results are evaluated on 8 Tesla V100-SXM2 GPUs with 32GB memory. Self-supervised pretraining for V-Dict-AE is run on 4 H100 GPUs. Database-side representations are computed once and cached, so evaluation is dominated by query-time dictionary extraction and nearest-neighbor search.

**Retrieval text encoder.** Dictionary serializations are embedded with OpenCLIP pretrained weights. Unless otherwise stated, the default text encoder is ViT-B/32, and Table 4 includes an ablation against a larger ViT-L/14 text encoder under the same retrieval protocol. The same text encoder is used for both indexing candidate dictionaries and embedding query dictionaries, ensuring a unified similarity space.

**V-Dict-AE pretraining setup.** V-Dict-AE is pretrained for 200 epochs using only images, without CIR triplets. The encoder backbone is Qwen2.5-VL-7B-Instruct (Bai et al., 2025), and the reconstruction signal is provided by a frozen Stable-Diffusion-2-base decoder. Two corpora are used to balance domain coverage: COCO2017 (Lin et al., 2014) for open-domain imagery and FashionAI (Zou et al., 2019) for fashion-specific appearance patterns. Optimization uses a learning rate of $5 \times 10^{-5}$, weight decay 0.01, and Adafactor with $\beta_1 = 0.9$.

**Qwen2.5-VL backbone and trained modules.** Qwen2.5-VL follows a standard multimodal decoder-only design: a visual encoder produces patch-level visual tokens, a projection or connector maps visual tokens into the language-model hidden space, and the language model performs joint multimodal reasoning and generation over the mixed sequence (Bai et al., 2025; Wang et al., 2024c). In Pix2Key, the frozen Qwen2.5-VL backbone is used for two purposes: extracting open-vocabulary key–value evidence from images and parsing edit intent into structured constraints. During V-Dict-AE pretraining, the diffusion model and its text interface remain frozen; training updates only lightweight components that shape the dictionary representation, including the attention pooler that compresses patch tokens into fixed-length dictionary slots, the projection head that aligns slots with the diffusion text-conditioning space, and optional low-rank adapters inserted into selected Qwen2.5-VL transformer layers. At inference time, the learned slot module augments the dictionary extractor, effectively replacing a generic captioner-style bottleneck with pretrained dictionary slots that better preserve fine-grained visual evidence while keeping the same retrieval interface.

**Baselines** Pic2Word (Saito et al., 2023) maps the reference image to a pseudo-word inserted into the text encoder input, so the composed query is formed by token-level composition in a frozen CLIP-style text space. PALAVRA (Cohen et al., 2022) learns new word embeddings from images to personalize a frozen vision–language space, providing an inversion-style mechanism that turns visual evidence into text tokens for retrieval. SEARLE (Baldrati et al., 2023) formulates composed retrieval via textual inversion, optimizing pseudo-tokens so that the composed query embedding better matches the intended target in a frozen retrieval space. FTI4CIR (Zhang et al., 2024) improves inversion-based composition with a fine-grained tokenization strategy, aiming to better capture subtle attribute changes while reducing reliance on curated CIR triplets. CIReVL (Karthik et al., 2024) uses a training-free language-space pipeline: a vision–language model produces textual descriptions and an LLM rewrites them under the edit, after which retrieval is performed by matching rewritten text to candidates in a pretrained embedding space. Context-I2W (Tang et al., 2024) makes the image-to-word mapping explicitly depend on the edit context and uses view selection to stabilize the mapping under clutter or viewpoint changes.

**Benchmarks.** FashionIQ (Wu et al., 2021) evaluates fashion CIR with relative language feedback. Results are reported on Dresses, Shirts, and Tops&Tees with Recall@10 and Recall@50 following the standard protocol. CIRR (Liu et al., 2021) evaluates open-domain composed retrieval on real-life images with natural-language edits. We report Recall@K under the standard candidate pool setup.

# B. DFMM-Compose: construction and processing

**Motivation.** Existing CIR benchmarks typically provide reference–target pairs and an edit description, but do not provide structured attributes for evaluating whether non-target candidates also satisfy the intended constraints. Conversely, attribute-centric fashion datasets provide labels but lack natural-language edits, preventing language-conditioned intent analysis. DFMM-Compose is introduced to bridge this mismatch by combining language edits with attribute-grounded evaluation.

| Auxiliary tag | ID $\rightarrow$ label mapping |
|---|---|
| `upper_color`/`lower_color`/`outer_color` | 0: black, 1: white, 2: grey, 3: beige, 4: brown, 5: red, 6: orange, 7: yellow, 8: green, 9: olive, 10: blue, 11: navy, 12: purple, 13: pink, 14: multicolor, 15: NA |
| `sleeve_style` | 0: regular set-in, 1: raglan, 2: drop-shoulder, 3: puff, 4: bell, 5: cap sleeve, 6: kimono, 7: dolman, 8: rolled, 9: cuffed, 10: flared, 11: NA |
| `upper_fit_silhouette` | 0: slim, 1: regular, 2: relaxed, 3: oversized, 4: crop-fitted, 5: boxy, 6: tunic, 7: longline, 8: bodycon, 9: NA |
| `lower_fit` | 0: skinny, 1: slim, 2: straight, 3: regular, 4: relaxed, 5: tapered, 6: wide-leg, 7: palazzo, 8: flared, 9: bootcut, 10: NA |
| `occasion_type` | 0: casual, 1: work, 2: formal, 3: party, 4: sport, 5: outdoor, 6: homewear, 7: NA |
| `seasonality` | 0: all-season, 1: spring, 2: summer, 3: fall, 4: winter, 5: spring–summer, 6: fall–winter, 7: NA |

*Table 5.* Auxiliary tag vocabularies (attribute maps) used for diversity evaluation in DFMM-Compose.

**Source data.** DFMM-Compose is built on the women split of DeepFashion-MM (Jiang et al., 2022), where each image is associated with 18 attribute labels. The candidate pool and attribute annotations are preserved to support attribute-based analysis of retrieved lists.

**Triplet construction.** Approximately 4,500 composed triplets are constructed from women's fashion images to match the intended evaluation domain. Each triplet defines a composed query and a corresponding target instance, while the full gallery remains available for ranking-based evaluation.

**Edit generation.** For each triplet, an edit description is generated from attribute differences between the reference and the target. GPT-4o is used to produce natural language modifications that are explicit about the intended attribute changes and consistent with the attribute labels. This yields queries that include both a reference image and a human-readable edit instruction.

**Auxiliary tags for diversity.** To quantify diversity beyond the original attribute vocabulary, eight auxiliary categorical tags are added to capture additional appearance factors that influence perceived variety: `upper_color`, `lower_color`, `outer_color`, `sleeve_style`, `upper_fit_silhouette`, `lower_fit`, `occasion_type`, and `seasonality`. Each tag is represented as a discrete ID and mapped to a human-readable string for evaluation (Table 5). The special value `NA` indicates that the tag is not applicable or missing for a given image. These tags are used only for evaluation and do not alter the retrieval pipeline.

**Evaluation tasks and metrics.** DFMM-Compose supports standard Recall@K evaluation and list-level evaluation over the top-ranked results. Attribute Consistency at top-50, AC@50, measures how frequently retrieved candidates satisfy the intended attribute constraints implied by the edit. Intra-List Diversity at top-50, ILD@50, measures redundancy within the returned list by aggregating pairwise distances in the attribute-and-tag space. This combination makes it possible to evaluate both intent satisfaction and user-facing diversity beyond a single labeled target.

## C. Distance-form MMR Reranking

**Algorithm 1** Distance-form MMR reranking used in Pix2Key

**Require:** Candidate pool $\mathcal{C}$, relevance score $R(i)$, distance $\mathrm{dist}(\cdot, \cdot)$, tradeoff $\lambda$, output size $K$
**Ensure:** Ordered list $S$
0: $S \leftarrow \{\arg\max_{i \in \mathcal{C}} R(i)\}$
0: **while** $|S| < K$ **do**
0: $\quad i^\star \leftarrow \arg\max_{i \in \mathcal{C} \setminus S} \left[ (1 - \lambda) R(i) + \lambda \min_{j \in S} \mathrm{dist}(i, j) \right]$
0: $\quad S \leftarrow S \cup \{i^\star\}$
0: **end while**
0: **return** $S$ =0

## D. Metric Definitions

**Notation.** For each query $q$, let the retrieved ranked list be $\pi_q = [i_1, i_2, \dots]$, where $i_k$ is the index of the $k$-th retrieved database image. For DFMM-Compose, each image is associated with (i) the original 18 DeepFashion-MM attribute labels and (ii) 8 auxiliary diversity tags, forming an extended attribute vector.

**Recall@K.** Recall@K is computed under the standard CIR protocol with a single annotated target image $i_q^\star$ for each query $q$.

$$\text{R@K} = \frac{100}{|\mathcal{Q}|} \sum_{q \in \mathcal{Q}} \mathbb{I}\big[i_q^\star \in \{i_1, \ldots, i_K\}\big], \tag{28}$$

where $\mathcal{Q}$ is the set of evaluation queries and $\mathbb{I}[\cdot]$ is the indicator function.

**AC@50: Attribute Consistency at top-50.** AC@50 measures intent satisfaction beyond retrieving a single labeled target by checking whether *each retrieved candidate* matches the attribute changes implied by the edit. Each DFMM-Compose query is constructed from a reference image $I_q^{\text{ref}}$ and a target image $I_q^{\text{tgt}}$ with 18 attribute labels. Let $\mathbf{a}_q^{\text{ref}} \in \mathcal{A}^{18}$ and $\mathbf{a}_q^{\text{tgt}} \in \mathcal{A}^{18}$ denote the corresponding attribute label vectors. The edited-attribute set is defined by label differences:

$$\mathcal{E}(q) = \{m \in \{1, \ldots, 18\} \mid \mathbf{a}_q^{\text{ref}}[m] \neq \mathbf{a}_q^{\text{tgt}}[m]\}. \tag{29}$$

For a retrieved candidate image $I_i$ with label vector $\mathbf{a}_i$, its per-candidate consistency score is

$$c(q, i) = \frac{1}{|\mathcal{E}(q)|} \sum_{m \in \mathcal{E}(q)} \mathbb{I}\big[\mathbf{a}_i[m] = \mathbf{a}_q^{\text{tgt}}[m]\big]. \tag{30}$$

AC@50 first averages this score over the top-50 results and then over queries:

$$\text{AC@50} = \frac{100}{|\mathcal{Q}|} \sum_{q \in \mathcal{Q}} \left( \frac{1}{50} \sum_{k=1}^{50} c(q, i_k) \right). \tag{31}$$

This definition focuses on *edited* attributes, so candidates can vary in other aspects without being penalized, which is aligned with user-facing CIR where multiple valid completions exist.

**ILD@50: Intra-List Diversity at top-50.** ILD@50 quantifies redundancy among the top-50 retrieved candidates using an attribute-based distance. Let $\mathbf{z}_i \in \mathcal{Z}^D$ denote the extended discrete descriptor of image $I_i$, where $D = 26$ includes the 18 original attributes plus 8 auxiliary diversity tags. For two images $I_i$ and $I_j$, define a normalized Hamming distance in the extended attribute space:

$$d_{\text{attr}}(i, j) = \frac{1}{D} \sum_{m=1}^{D} \mathbb{I}\big[\mathbf{z}_i[m] \neq \mathbf{z}_j[m]\big]. \tag{32}$$

ILD@50 averages pairwise distances within the top-50 list:

$$\text{ILD@50} = \frac{100}{|\mathcal{Q}|} \sum_{q \in \mathcal{Q}} \left( \frac{2}{50 \cdot 49} \sum_{1 \leq u < v \leq 50} d_{\text{attr}}(i_u, i_v) \right). \tag{33}$$

Unlike embedding-space diversity, this attribute-based formulation explicitly measures whether the returned candidates differ in *interpretable* appearance factors, and the 8 auxiliary tags provide additional resolution beyond the original 18 labels.

### D.1. Additional Robustness and Generalization

**Robustness to vague edit instructions.** To evaluate robustness under noisier user inputs, we construct a harder FashionIQ setting by injecting vague instructions into the original relative captions, such as "I want a more beautiful dress." As shown in Table 6, all compared baselines suffer clear performance drops, while Pix2Key+V-Dict-AE remains much more stable. This suggests that the dictionary representation is less sensitive to underspecified or colloquial edit text, since explicit desired, forbidden, and open attributes help separate the intended edit from weakly informative language.

**Generalization to unseen categories and shifted domains.** We further evaluate whether Pix2Key generalizes beyond the main FashionIQ category space. First, we test on a FACap-skirt subset, where skirt is not part of the original FashionIQ categories. Second, we construct FashionIQ-CustomerShow by replacing part of the original FashionIQ targets with generated buyer-show or try-on images, while keeping the same reference image and relative caption. As shown in Table 7, Pix2Key+V-Dict-AE consistently outperforms strong baselines on the unseen category and degrades less under domain shift.

| Method | Original Avg | Vague Avg | Drop |
|---|---|---|---|
| CIReVL | 27.87 | 24.93 | -2.94 |
| Pic2Word | 24.70 | 21.58 | -3.12 |
| SEARLE | 25.56 | 22.10 | -3.46 |
| FTI4CIR | 29.39 | 25.61 | -3.78 |
| Pix2Key+V-Dict-AE | **29.96** | **29.28** | **-0.68** |

*Table 6.* Robustness to vague edit instructions on FashionIQ. Pix2Key+V-Dict-AE shows a much smaller performance drop than prior baselines.

| Method | FACap R@10 | FACap R@50 | Dresses R@50 | Shirts R@50 | Tops&Tees R@50 | Avg R@50 | Avg Drop |
|---|---|---|---|---|---|---|---|
| CIReVL | 24.8 | 46.3 | 41.31 | 43.47 | 48.65 | 44.48 | -4.20 |
| FTI4CIR | 25.6 | 47.2 | 43.40 | 46.23 | 49.90 | 46.51 | -4.37 |
| Pix2Key+V-Dict-AE | **27.8** | **50.4** | **47.18** | **49.81** | **53.87** | **50.29** | **-1.62** |

*Table 7.* Generalization analysis. Left: unseen-category evaluation on FACap-skirt. Right: domain-shift evaluation on FashionIQ-CustomerShow.

**Query-time efficiency.** We compare query-time cost with the closest training-free caption-rewrite baseline under the same Qwen2.5-VL backbone. As shown in Table 8, Pix2Key is slightly faster and uses less peak GPU memory than CIReVL while achieving stronger retrieval performance. Using a smaller Qwen2.5-VL-3B backbone further reduces latency and memory with only a small performance drop, suggesting that the dictionary interface is compatible with more efficient deployment settings.

| Method | Backbone | Latency ↓ s/query | Memory ↓ GB | R@50 ↑ | AC@50 ↑ |
|---|---|---|---|---|---|
| CIReVL | Qwen2.5-VL-7B | 1.62 | 19.3 | 35.40 | 36.42 |
| Pix2Key | Qwen2.5-VL-7B | 1.28 | 17.9 | 37.56 | 51.26 |
| Pix2Key | Qwen2.5-VL-3B | **0.82** | **11.8** | 36.94 | 50.31 |

*Table 8.* Query-time efficiency comparison. Pix2Key improves both retrieval quality and efficiency over CIReVL under the same VLM backbone, and remains stable with a smaller VLM.

