# OpenReview forum: "Pix2Key: Controllable Open-Vocabulary Retrieval with Semantic Decomposition and Self-Supervised Visual Dictionary Learning"
_ICML.cc/2026/Conference — ICML 2026 regular_

### Official Review · Reviewer_PeU7 · 2026-03-10

**Soundness:** 3
**Presentation:** 3
**Significance:** 2
**Originality:** 3
**Overall Recommendation:** 4
**Confidence:** 4

**Summary:**

This paper studies controllable open-vocabulary generation/recognition and proposes a framework (Pix2Key) that aims to bridge pixel-level visual evidence with a more structured “key” representation to enable controllability. The core idea is to provide an explicit handle (the “key”) so that the model’s outputs can be guided by user intent while still supporting open-vocabulary generalization. The paper positions the approach as a practical way to improve control compared with purely prompt-based or implicit control methods, and presents experiments to demonstrate effectiveness under different control settings.

**Compliance With Llm Reviewing Policy:**

Affirmed.

**Final Justification:**

Thank you to all the authors for your responses. My doubts have been resolved. It is suggested that the authors incorporate these revised content into the final version.

**Key Questions For Authors:**

1. What are the exact user-controllable inputs (“keys”) in this framework, and how sensitive is performance to imperfect or noisy keys?

2. Does the approach require any task-specific annotations for keys, or can keys be obtained automatically from existing signals?

3. How does Pix2Key behave when the control signal conflicts with the image evidence (or when multiple controls disagree)?

**Limitations:**

The authors do not provide limitations. Maybe need more discussion on critical technical limitations (see the questions).

**Strengths And Weaknesses:**

Strengths

1. The topic is timely and relevant: controllable open-vocabulary modeling is important for practical usage and evaluation.

2. The high-level motivation is clear: introducing an explicit intermediate handle for control is intuitive and aligns with how users want to specify constraints.

3. The method appears modular and potentially compatible with existing backbones/pipelines, which is a plus for adoption.
The paper focuses on a concrete controllability goal rather than only reporting generic performance gains, which is appreciated for a conference submission.

Weaknesses

1. Positioning against closest baselines could be clearer. The paper would benefit from a short, direct explanation of what Pix2Key changes compared with the most relevant controllable/open-vocabulary baselines (e.g., what is controlled, at what stage, and what supervision or signals are required).

2. Control specification and failure modes are not fully characterized. It would help to briefly summarize what kinds of controls are supported (and not supported) and provide a small set of typical failure cases (e.g., ambiguous control, conflicting constraints, unseen concepts), so readers understand the practical boundary of the method.

3. Efficiency/overhead is not discussed. If Pix2Key introduces additional components or extra inference steps, a short note on runtime or relative overhead would make the evaluation more complete and help judge deployability.

4. Evaluation coverage could be strengthened with a simple additional check. For example, adding one more setting that tests generalization (new categories, different domains, or a more challenging open-vocabulary split) or reporting results under a second backbone would better support the “open-vocabulary” claim without requiring major new experiments.

---

> ### Author Rebuttal · Authors · 2026-03-31
>
> **Q1:Differences from baselines**
>
> **A1:**  We summarize the main differences between Pix2Key and the closest baselines in the table below. The core idea of **Pix2Key** is to represent the **reference image** as a **structured dictionary**, where fine-grained attributes are made explicit. In contrast, the closest baselines typically represent the image as **latent vectors** or a **single caption**. As a result, when an edit request is given (e.g., changing a blue skirt to **red**), Pix2Key can directly update the corresponding attribute in the dictionary, making the control signal more explicit and interpretable. This structured form can also be directly consumed by the language model, without requiring in-domain supervision.
>
> | Method | Image representation format | Direct interaction with edit prompt (controllability) | Need image-text pair supervision |
> |-|-|-|-|
> | Pic2Word | a latent vector | No | Yes |
> | PALAVRA | a latent vector | No | Yes |
> | SEARLE | a latent vector | No | Yes |
> | FTI4CIR | multiple latent vectors | No | Yes |
> | CIReVL | a caption | Partial | No |
> | **Pix2Key (ours)** | a structured dictionary | Yes | No |
>
> **Q2: Control specification and failure modes**
>
> **A2:** Pix2Key provides **semantic attribute-level control** by modeling which attributes should be promoted, avoided, or left open. This supports both fine-grained appearance edits and more general composed retrieval.
>
> Typical failure cases include (i) **conflicting constraints**, where requested properties are incompatible, and (ii) **unseen or weakly grounded concepts**, where the needed knowledge is absent from the underlying VLM/backbone. Therefore, Pix2Key supports **semantic-level control**, but not very fine-grained coordinate- or region-level control. We will clarify these supported controls, failure cases, and practical boundaries in the revision.
>
> **Q3: Efficiency**
>
> **A3:** Baselines such as **Pic2Word, PALAVRA, SEARLE, and FTI4CIR** require in-domain partial finetuning or distillation of their embedding model, i.e., **CLIP**, which accounts for a substantial part of the computation cost and is not directly comparable to our framework.
>
> We therefore provide a direct efficiency comparison with the closer baseline **CIReVL** below. Under the same default VLM backbone (**Qwen2.5-VL-7B**), Pix2Key is slightly better in both query latency and peak GPU memory, while achieving much stronger retrieval performance. Replacing the default backbone with a smaller VLM (**Qwen2.5-VL-3B**) yields a substantial efficiency gain with only a very small performance drop. This suggests that our dictionary-based representation is robust and can likely support even smaller backbones for more efficient deployment.
>
> | Method | Backbone | Query latency (s/query) ↓ | Peak GPU memory (GB) ↓ | R@50 ↑ | AC@50 ↑ |
> |---|---|---:|---:|---:|---:|
> | CIReVL | Qwen2.5-VL-7B |1.62|19.3|35.40|36.42|
> | Pix2Key | Qwen2.5-VL-7B |1.28|17.9|37.56|51.26|
> | Pix2Key | Qwen2.5-VL-3B |0.82|11.8|36.94|50.31|
>
> **Q4: generalization results.**
>
> See Reviewer JH17 Q3.
>
> **Q5. What are the controllable keys?**
>
> **A5:** In Pix2Key, the controllable “keys” are semantic attribute keys extracted by a VLM from the query image and the user’s edit prompt. According to user intent, they are divided into three types: **desired**, **forbidden**, and **open** attributes. During ranking, we compute the corresponding scores for these three types to control retrieval results. More details can be found in **Reviewer TU1z, Q1**.
>
> **Q6. Noise sensitivity**
>
> See **Reviewer TU1z, Q1/Q2** and **Reviewer JH17, Q3**.
>
> **Q7: Key extraction and annotation requirements**
>
> **A7:** Pix2Key does not require task-specific annotations for keys. Instead, the keys are obtained automatically from existing signals: a VLM extracts key–value evidence from the reference image, and the natural-language edit is parsed into desired / forbidden / open constraints, which together form the signed query dictionary. Our framework is therefore training-free at retrieval time, and even the V-Dict-AE component is trained self-supervised using only images; structured attributes in DFMM-Compose are used only for evaluation.
>
> **Q8. Conflicting control signals**
>
> **A8:** When the control signal conflicts with the image evidence, Pix2Key prioritizes the user’s edit and preserves only compatible reference context. Specifically, edited attributes override conflicting reference attributes on the same key, while open attributes act only as a **soft preservation prior**. Thus, the reference image is not treated as a hard constraint.
>
> When multiple controls disagree, the conflict remains explicit at the **key level**, making the behavior more interpretable than in a single fused embedding. A remaining limitation is truly contradictory instructions on the same key, where the final behavior depends on the parsed dictionary and merge result. We will clarify this failure mode in the revision.

---

> > ### Author Rebuttal · Reviewer_PeU7 · 2026-04-03
> >
> > I would like to thank the authors for their detailed rebuttal, which has addressed some of my initial concerns. However, several critical points remain unclear, and I would appreciate further clarification on the following issues:
> >
> > 1. How does the framework handle semantic inconsistencies during the dictionary extraction phase? If the text encoder $f_{text}$ (e.g., OpenCLIP) maps key-value pairs that are semantically similar but syntactically different to distant points in the embedding space, would this significantly degrade retrieval precision?
> >
> > 2. While the experiments demonstrate that reconstruction-based pretraining improves Recall, is "visual saliency" for reconstruction truly equivalent to the "attribute discriminability" required for retrieval? Are there certain noisy features that are vital for image reconstruction but may interfere with the ability to distinguish between highly similar garments?

---

> > > ### Author Response · Authors · 2026-04-04
> > >
> > > Dear Reviewer PeU7,
> > >
> > > We sincerely appreciate your thoughtful follow-up questions and the time you have spent on our paper. We have conducted additional experiments to address your remaining concerns, and we clarify them below.
> > >
> > > **1. Semantic inconsistencies in dictionary extraction / serialization.**
> > >
> > > We verify that our text encoder \(f_{\text{text}}\) produces much closer embeddings for expressions with the same semantics but different syntactic forms, which suggests that syntactic variation has only a very limited effect on retrieval. We attribute this to CLIP’s strong robustness to diverse expressions and good sensitivity informative semantics, which is consistent with its contrastive pretraining objective.
> > >
> > > We first test CLIP on controlled sentence pairs such as:
> > >
> > > - **S1** (Template A + Clothing X): “A photo of a red dress”
> > > - **S2** (Template B + Clothing X): “Someone is wearing a red dress”
> > > - **S3** (Template B + Clothing Y): “Someone is wearing a black jacket”
> > >
> > > We encode 1,000 such cases and obtain the following average cosine similarities:
> > >
> > > | Pair | Relation | Cosine Similarity |
> > > |------|----------|:-:|
> > > | S1 vs S2 | Same clothing, different template | 0.9122 |
> > > | S2 vs S3 | Same template, different clothing | 0.6854 |
> > > | S1 vs S3 | Different template, different clothing | 0.6047 |
> > >
> > > Here, **S1** and **S2** share only the key semantic content (e.g., “red dress”), while the surrounding expression is quite different, yet their similarity is above 0.9, which is already a very high score for retrieval. In contrast, pairs with the same template but different semantic content have substantially lower similarity. This suggests that the encoder is much more sensitive to semantic changes than to template-level wording differences.
> > >
> > > We further test this at the retrieval level on DFMM-Compose. Specifically, we randomly replace the dictionary keys in our candidate pool using LLM-generated synonyms (Claude Opus 4.6), and evaluate retrieval again. The results are shown below:
> > >
> > > | Method | AC@50 ↑ | ILD@50 ↑ | R@10 ↑ | R@50 ↑ |
> > > |---|---:|---:|---:|---:|
> > > | Pix2Key+V-Dict-AE original | 54.44 | 53.96 | 23.58 | 40.96 |
> > > | Pix2Key+V-Dict-AE rewritten | 54.32 | 53.88 | 23.41 | 40.83 |
> > >
> > > The performance change is very small, which further supports that moderate lexical variation in dictionary expressions does not materially affect retrieval in practice.
> > >
> > > **2. Reconstruction saliency vs. retrieval discriminability.**
> > >
> > > Thank you for raising this important concern. The key connection between “visual saliency” and “attribute discriminability” in our method lies in the **representation format**. A core design choice of Pix2Key is exactly to create an information bottleneck through the dictionary representation.
> > >
> > > During training, V-Dict-AE is required to encode the image into a **limited number of attributes** and reconstruct the input from these attributes. This means that the extracted attributes must be a highly distilled summary of the input. In contrast, noisy features that may help reconstruct low-semantic image details are less likely to appear in the dictionary, because they are not important enough to survive this bottleneck.
> > >
> > > We further support this claim with the following experiment. We retrain V-Dict-AE with wider intermediate representations by replacing the dictionary bottleneck with either (i) fully free-form natural language, or (ii) latent language tokens. In both cases, the information bottleneck is loosened. We then evaluate retrieval performance on DFMM-Compose:
> > >
> > > | Representation format | Information bottleneck | AC@50 ↑ | ILD@50 ↑ | R@10 ↑ | R@50 ↑ |
> > > |---|---|---:|---:|---:|---:|
> > > | Dictionary (our default) | narrow | 54.44 | 53.96 | 23.58 | 40.96 |
> > > | Free-form natural language | wide | 51.02 | 53.71 | 20.18 | 37.72 |
> > > | Latent language tokens | wider | 49.38 | 53.12 | 18.66 | 36.95 |
> > >
> > > We observe a consistent degradation in retrieval performance as the bottleneck becomes wider, with the clearest drop appearing in AC@50, which more directly reflects attribute discriminability. This supports our claim that the narrower dictionary bottleneck helps preserve retrieval-relevant semantic attributes, rather than redundant reconstruction-oriented details.
> > >
> > > We hope these additional experiments clarify the remaining concerns, and we sincerely appreciate your helpful questions.

---

### Official Review · Reviewer_JH17 · 2026-03-11

**Soundness:** 3
**Presentation:** 3
**Significance:** 3
**Originality:** 3
**Overall Recommendation:** 4
**Confidence:** 3

**Summary:**

The submission proposes Pix2Key, a training-free composed image retrieval framework that converts both query and gallery images into open-vocabulary key–value visual dictionaries, and decomposes the edit into polarity constraints: desired, avoided, and open-set anchors. Dictionaries are serialized and embedded with a frozen OpenCLIP text encoder for unified text-space retrieval, followed by MMR-based reranking to balance relevance and diversity. To strengthen fine-grained attributes without CIR triplets, V-Dict-AE is introduced: a self-supervised visual-dictionary autoencoder, which learn slot tokens aligned to a frozen diffusion model’s text interface via reconstruction losses while training only lightweight modules. Experiments on FashionIQ, CIRR, and the new DFMM-Compose benchmark show consistent gains in Recall, attribute consistency (AC@50), and intra-list diversity (ILD@50).

**Compliance With Llm Reviewing Policy:**

Affirmed.

**Final Justification:**

Thanks the authors for the detailed rebuttal. My concerns are addressed with evidences, like the robustness and broader generalization. I tend to accept this submission.

**Key Questions For Authors:**

1. How robust are the dictionary extraction to domain shift and noisy/ambiguous edits, and can uncertainty calibration or light supervision improve reliability?

2. Does multi-embedding or per-key indexing outperform single global text embeddings for localized attributes without hurting efficiency?

3. How do relevance–diversity trade-offs (λ) and polarity weights (α, β) generalize across tasks, and can they be auto-tuned per query?

**Limitations:**

None. I have discussed them in the 'weakness' part.

**Strengths And Weaknesses:**

Strengths

1. Clear controllability via polarity-decomposed visual dictionaries (desired, avoided, or open anchors) enable explicit, interpretable constraint handling.

2. Training-free retrieval pipeline with offline indexing in a unified text space (OpenCLIP) is simple to deploy, and is efficient at scale.

3. Diversity-aware reranking (MMR distance form) improves result variety, without noticeably sacrificing relevance.

4. V-Dict-AE self-supervision strengthen fine-grained attribute fidelity using only images, training lightweight modules (pooler, projection or LoRA).

5. Consistent improvements on FashionIQ, CIRR, and DFMM-Compose across Recall, AC@50, and ILD@50 demonstrate robustness.


Weaknesses

1. Reliance on VLM-based dictionary, extraction and prompt quality may introduce variability across domains or noisy attributes.

2. Single text embedding per candidate can be a bottleneck for highly compositional or localized attributes. Multi-embedding indexing might help.

3. Ontolog or serialization choices for key–value dictionaries may limit coverage or consistency. Hierarchical or graph-structured schemas could be beneficial.

4. Limited discussion of latency and scalability trade-offs for very large galleries (e.g., ANN settings, candidate pool sizes, λ selection).

5. Potential bias and domain shift risks from frozen backbones and diffusion priors. Broader fairness or robustness evaluations are desirable.

6. Open anchors can dilute attribute specificity if not paired with strong negative constraints, which requires careful hyperparameter tuning (α, β, λ).

7. Evaluation focuses on image domains similar to fashion or open-domain datasets. I think more tests on cluttered scenes, small objects, and occlusions would strengthen claims.

---

> ### Author Rebuttal · Authors · 2026-03-31
>
> We sincerely thank the reviewer for the thoughtful and constructive feedback, and for recognizing the controllability, efficiency, and consistent empirical gains of Pix2Key.
>
> **Q1. Robustness to domain shift / noisy edits, and possible ways to improve reliability**
>
> **A1:**. As discussed in Reviewer PeU7 Q5, the controllable “keys” in Pix2Key are polarity-aware semantic key-value constraints parsed from the reference image and edit text, namely desired, forbidden, and open attributes. The open term acts as a soft preservation prior, helping preserve compatible reference-side context without competing with the intended edit. In addition, as shown in Reviewer PeU7 Q4, Pix2Key+V-Dict-AE remains strong on the unseen FACap-skirt category and degrades much less than prior baselines on the FashionIQ-CustomerShow domain-shift benchmark, supporting robustness beyond the original category space and image style. We also note that V-Dict-AE improves reliability in a fully self-supervised way, without requiring extra image-caption pairs or CIR triplets. Uncertainty calibration, light supervision, and broader bias/fairness analysis are promising future directions that we plan to explore.
>
> **Q2. Multi-embedding vs. single-embedding indexing**
>
> **A2:** Our current design already achieves a strong efficiency–performance trade-off. Specifically, gallery items are indexed by a single global embedding for efficient retrieval, but the query itself is not reduced to a single vector: it is decomposed into three embeddings (positive / negative / open), which are scored separately before aggregation. This structured multi-channel query design already captures more compositional information while preserving efficient candidate indexing. This is also consistent with Table 3, where performance improves steadily under the same single-embedding gallery setup (R@50: 37.27 → 39.41 → 40.96; AC@50: 52.49 → 53.31 → 54.44). We therefore believe the current design already alleviates much of the candidate-side single-embedding bottleneck while keeping the system scalable. Multi-embedding or per-key indexing is an interesting future direction, but it would introduce additional indexing and retrieval overhead. Related efficiency evidence is also provided in Reviewer PeU7 Q3.
>
> **Q3. Broader generalization.**
>
> **A3:** We thank the reviewer for the helpful suggestion. To further evaluate the generalization ability of Pix2Key, we provide additional evidence from three perspectives:
>
> (1) **Real-world generalization.** The text queries in all three datasets are already written in natural language and are close to real-world retrieval scenarios. To further stress this setting, we build a harder FashionIQ benchmark by injecting additional vague instructions into each original query, such as “I want a more beautiful dress.” As shown below, all baselines show a clear drop, while Pix2Key+V-Dict-AE remains almost unchanged, demonstrating strong robustness to vague and noisy user inputs.
>
> | Method | Original Avg | Vague Avg | Drop |
> |---|---:|---:|---:|
> | CIReVL | 27.87 | 24.93 | -2.94 |
> | Pic2Word | 24.70 | 21.58 | -3.12 |
> | SEARLE | 25.56 | 22.10 | -3.46 |
> | FTI4CIR | 29.39 | 25.61 | -3.78 |
> | Pix2Key+V-Dict-AE | **29.96** | **29.28** | **-0.68** |
>
>
> (2) **Generalization to new categories.** We evaluate on a FACap-**skirt** subset with about 2k triplets, where **skirt** lies outside the category space of our main FashionIQ benchmark. Pix2Key+V-Dict-AE consistently outperforms strong baselines on this unseen category.
>
> | Method | FACap-skirt R@10 ↑ | FACap-skirt R@50 ↑ |
> |---|---:|---:|
> | CIReVL | 24.8 | 46.3 |
> | FTI4CIR | 25.6 | 47.2 |
> | Pix2Key+V-Dict-AE | **27.8** | **50.4** |
>
> (3) **Robustness under domain shift.** We construct FashionIQ-CustomerShow by partially replacing the original FashionIQ test targets with Nano Banana generated buyer-show / try-on images, while keeping the reference image and relative caption unchanged. As shown below, Pix2Key+V-Dict-AE degrades much less than prior methods, showing strong robustness under realistic domain shift.
>
> | Method | Dresses R@50 ↑ | Shirts R@50 ↑ | Tops&Tees R@50 ↑ | Avg R@50 ↑ | Avg drop ↓ |
> |---|---:|---:|---:|---:|---:|
> | CIReVL | 41.31 | 43.47 | 48.65 | 44.48 | -4.20 |
> | FTI4CIR | 43.40 | 46.23 | 49.90 | 46.51 | -4.37 |
> | Pix2Key+V-Dict-AE | **47.18** | **49.81** | **53.87** | **50.29** | **-1.62** |
>
> **Q4  Efficiency and scalability**
>
> **A4:**   Compared with CIReVL under the same **Qwen2.5-VL-7B** backbone, Pix2Key achieves lower latency (**1.62s → 1.28s**) while delivering stronger retrieval performance (**R@50: 35.40 → 37.56**). It also remains robust when switching to a much smaller VLM; see **Reviewer PeU7, Q3** for details.
>
> **Q5. Reliability of dictionary extraction**
>
> **A5:** Please refer to Reviewer TU1z Q3.
>
> **Q6. Stability of α, β, λ and open anchors**
>
> **A6:** Please refer to Reviewer TU1z Q1 and Q2.

---

> > ### Author Rebuttal · Reviewer_JH17 · 2026-04-03
> >
> > Thanks the authors for the responses in detail. My concerns are addressed with evidences, like the robustness and broader generalization. I tend to accept this submission. While I'm also open to further discussion with other reviewers.

---

> > > ### Author Response · Authors · 2026-04-04
> > >
> > > Dear Reviewer JH17,
> > >
> > > Thank you very much for your thoughtful review and especially for your encouraging and positive feedback. We truly appreciate your recognition of the controllability, efficiency, and consistent empirical improvements of our method. Your support means a lot to us. We are also grateful for your helpful suggestions, which guided us to further strengthen the analysis and improve the clarity of the paper. Thank you again for your time and valuable feedback.
> > >
> > > Best regards,
> > > Authors

---

### Official Review · Reviewer_Mzdu · 2026-03-12

**Soundness:** 3
**Presentation:** 3
**Significance:** 2
**Originality:** 2
**Overall Recommendation:** 3
**Confidence:** 4

**Summary:**

This paper addresses the issues of insufficient fine-grained attribute capture, homogeneous retrieval results on supervised triple training in the zero-shot combined image retrieval (CIR) task. It proposes the Pix2Key framework. This framework uniformly represents the query and candidate images as an open vocabulary visual dictionary, decomposing the user editing instructions into structured attribute constraints with polarity and combining diversity-aware reordering to balance the relevance and diversity of the retrieval results. Meanwhile, it proposes the self-supervised pre-training module V-Dict-AE, which only optimizes the dictionary representation through image data to enhance the understanding ability of fine-grained attributes. Additionally, the paper constructs the DFMM-Compose benchmark, which quantitatively evaluates the attribute consistency and list diversity of the retrieval results. Some comparative experiments on mainstream datasets such as FashionIQ and CIRR.

**Compliance With Llm Reviewing Policy:**

Affirmed.

**Key Questions For Authors:**

How do you formally justify the essential innovation and incremental academic contribution of the proposed visual dictionary framework compared with existing attribute-level structured retrieval methods, and how do you ensure the fairness of baseline comparisons to isolate the performance gains brought by your design? What is the generalization performance of the method in complex edit scenarios and noisy real-world inputs?

**Limitations:**

None.

**Strengths And Weaknesses:**

Strengths:
1. The paper focuses on the common industry pain points in zero-shot composed image retrieval tasks, including the loss of fine-grained editing information and homogeneity of retrieval results. The research direction aligns with practical application scenarios of cross-modal retrieval, and the topic has strong practical relevance.
2. The paper constructs the DFMM-Compose benchmark to supplement the evaluation of attribute-level intent satisfaction and list-level diversity, which fills the gap that existing CIR benchmarks only focus on single-target hit rate. The experimental evaluation covers multiple mainstream datasets, with complete ablation studies and hyperparameter sensitivity analysis.
3. The paper forms a complete technical narrative from problem formulation, method design to experimental verification, with a clear logical structure and standardized presentation of experimental data, which adheres to the basic writing conventions of academic papers.
Weaknesses:
1. The paper lacks sufficient theoretical justification and comparative analysis for the core visual dictionary design, and fails to clarify the essential difference between the proposed method and existing attribute-level structured retrieval works, resulting in a very limited incremental academic contribution of the core design.
2. Although the paper emphasizes the controllability of fine-grained edits, it does not verify the generalization performance of the method in complex real-world edit scenarios, such as vague instructions, multi-attribute linkage modifications and colloquial expressions. Meanwhile, the design of the edit decomposition and dictionary merging logic is overly simplistic, which cannot adapt to diverse user needs.
3. The diversity-aware reranking module directly adopts the classic MMR algorithm with only simple numerical normalization adaptation, and no targeted algorithm improvements are made for the CIR scenario and dictionary representation, resulting in no original contribution to this module.
4. The paper does not compare with stronger structured retrieval baselines and fairly aligned baselines, such as caption-rewrite methods with the same prompt optimization and attribute-level CIR methods with similar structured design, making it impossible to isolate the performance gains brought by the proposed framework itself.
5. The paper does not carry out any robustness experiments on real scenarios such as noisy edit instructions, fuzzy queries, and out-of-distribution data, which cannot support the claim that the method can solve the "CIR problem in real Internet data" as stated in the paper.

---

> ### Author Rebuttal · Authors · 2026-03-31
>
> **Q1. Contribution and novelty concerns.**
>
> **A1.** Thank you for your review. We respectfully disagree with you comments that our model makes incremental contribution over existing attribute-level structured retrieval works. To our knowledge, Pix2Key is the first approach that represents query information into a free-form dictionary for CIR. CIR is not a structured retrieval task --- most existing studies represent the query information into latent vectors and then perform retrieval, while this paper identifies their shortcomings of flexiblity and controllability. The core contribution of Pix2Key is that it innovatively encodes unstructured query information, including the reference image and editing prompts, into a structured dictionary representation. Such an explicit attribute-based formulation is naturally suited for retrieval tasks, and brings substantial performance improvements.
>
> **Theoratical justifications:** Our method can be viewed as a structured approximation to CIR attribute matching. Let $P$, $N$, and $O$ denote the desired, forbidden, and anchor attributes in the query. The ideal relevance is$$F^\star(I\mid q)=\sum_{a\in P} w_a x_a(I)-\sum_{a\in N} u_a x_a(I)+\sum_{a\in O} r_a x_a(I).$$Our dictionary representation scores these parts separately:$$R(I)=\alpha\langle q^+,e_{\mathrm{dict}}(I)\rangle+\beta\langle q^0,e_{\mathrm{dict}}(I)\rangle-(1-\alpha)\langle q^-,e_{\mathrm{dict}}(I)\rangle.$$This gives a more decomposable relevance signal, since different semantic factors are matched explicitly rather than being collapsed into one vector. By contrast, a single fused latent $z=A\,s(q)$ with $m<d$ is non-injective, so distinct fine-grained intents can collide:$$A\,s(q)=A\bigl(s(q)+\delta\bigr),\qquad \delta\ne 0.$$This is undesirable for CIR, where the edit is sparse but preserving the remaining identity details is crucial.
>
> **Q2. Real-world generalization performance**
>
> **A2:** The text queries in all datasets used in this paper are presented in natural language form, which is already quite close to real-world retrieval scenarios. The multi-attribute linkage modifications and colloquial expressions you mentioned are in fact well covered by all datasets. Here we construct a more challenging benchmark for FashionIQ, in which we inject vague instructions as noise into each original text query, such as “I want a more beautiful dress.” As shown in the table below, all baselines suffer from a clear performance drop, while Pix2Key's performance is almost unchanged. This demonstrates that **real-world generalization is not a weakness of our method; instead, it is a key strength.** This result also indicates that our simple edit decomposition and dictionary merging strategy is already sufficient to satisfy diverse user needs, without introducing more complicated designs.
> |Method|Original Avg|Vague Avg|Drop|
> |-|-|-|-|
> |CIReVL|27.87|24.93|2.94|
> |Pic2Word|24.70|21.58|3.12|
> |SEARLE|25.56|22.10|3.46|
> |FTI4CIR|29.39|25.61|3.78|
> |Pix2Key|29.96|29.48|0.48|
>
> **Q3: Contribution of reranking module**
>
> **A3**: Designing a more advanced reranking algorithm is not the main scope of this paper. Our contribution lies in identifying the clear deficiency of existing CIR models in terms of diversity, and in explicitly incorporating a design to address this issue into our framework. MMR algorithm is already sufficiently effective for standard retrieval settings, and therefore it is not necessary in this work to focus on improving this module.
>
> **Q4: Comparison to structured retrieval baselines**
>
> **A4**: We believe there may be a misunderstanding about the task setting. The task is retrieval conditioned on a reference image and a text query. It does not provide any structured data, and is more challenging than structured retrieval. Standard structured-retrieval baselines are not applicable to the task considered here. We have conducted extensive ablations in Table 3 and 4, isolating the effect of positive/negative/open embeddings, diversity-aware reranking, and prompt refinement, etc.
>
> The key distinction of Pix2Key is that it transforms unstructured queries into structured information that is more amenable to retrieval. Precisely because of this design, Pix2Key is better able to handle entangled semantics without being easily affected by noise, allowing it to consistently outperform existing models in realistic scenarios.
>
> **Q5. Robustness**
>
> **A5:** In A2, we show robustness to vague instructions, covering noisy edit requests and fuzzy queries. Here we provide robustness results on out-of-distribution and cross-domain settings in Reviewer PeU7's Q4. Under both settings, our method remains consistently stronger than the baselines, and under domain shift it shows substantially smaller performance degradation.
>
> **Additional question:** *We notice that the reviewer indicates CIR as "Combined Image Retrieval", yet we are working on "Composed Image Retrieval" in this paper. Is this indicating another task or a typo?*

---

> > ### Author Rebuttal · Reviewer_Mzdu · 2026-04-01
> >
> > The authors have addressed some of my concerns, but some of the doubts in Weaknesses remain unresolved.

---

> > > ### Author Response · Authors · 2026-04-01
> > >
> > > Thanks, but the current acknowledgement looks kind of vague and confusing. Can you let us know which part of the concerns is addressed and which part is not? We'd like to discuss more about your unaddressed problems.
> > >
> > > Additionally, can you confirm if the term "Combined Image Retrieval" is a typo or another task you are indicating? We believe there is a substantial misunderstanding in your review.

---

### Official Review · Reviewer_TU1z · 2026-03-12

**Soundness:** 3
**Presentation:** 3
**Significance:** 2
**Originality:** 3
**Overall Recommendation:** 4
**Confidence:** 4

**Summary:**

This paper focuses on Composed Image Retrieval (CIR), which uses a reference image plus text editing instructions to retrieve images that meet modification requirements while preserving other visual content. The authors point out that existing methods typically compress images and text into a single fused embedding, leading to the loss of fine-grained attribute information and producing repetitive and unsustainable retrieval results.

The authors propose Pix2Key, which represents images and queries as structured visual dictionaries (attribute key-value pairs) and parses text edits into attribute constraints, enabling retrieval through attribute-level matching. The system also incorporates diversity-aware reranking to improve result diversity and enhances the fine-grained expressiveness of the dictionary representation through V-Dict-AE self-supervised pre-training.

Traditional CIR methods rely on triplet supervision or fuse image and text into a single embedding, while zero-shot methods typically use tokens or captions to represent reference images. In contrast, Pix2Key uses explicit attribute dictionaries and constraint matching to model user intent, making the retrieval process more controllable and interpretable, and reducing reliance on specialized CIR annotation data.

**Compliance With Llm Reviewing Policy:**

Affirmed.

**Final Justification:**

Overall, the paper is technically sound with solid empirical performance. However, the theoretical motivation and conceptual insights remain somewhat limited, which may constrains the overall impact.

Therefore, I maintain my recommendation of Weak Accept.

**Key Questions For Authors:**

1. **Regarding the structural design of Eq. 9.**
   The scoring function couples the weights for $p_i$ ($p=+1$) and $n_i$ ($p=-1$) while treating $o_i$ ($p=0$) separately. In realistic scenarios, queries may contain only positive edits or only negative constraints, in which case the score could be dominated by the anchor term $o_i$. Although Table 3 provides ablation results, it is unclear why this potential structural issue does not appear to negatively affect performance. Could the authors provide further analysis explaining why the scoring formulation remains stable under such query distributions?

2. **Additional analysis for the components discussed in the weaknesses.**
   It would be helpful if the authors could provide additional experiments or analyses related to Eq. 9 and the intent extraction module. For example, (i) a sensitivity study of the hyperparameters $\alpha$ and $\beta$, (ii) experiments analyzing model behavior under queries containing only positive or only negative constraints, and (iii) evaluation of the accuracy or robustness of the Qwen-VL-based intent extraction, especially under more complex or diverse user queries.

**Limitations:**

NO.
1. The scoring function in Eq. 9 may have a potential design limitation. Since the weights for positive ($p=+1$) and negative ($p=-1$) constraints are coupled while anchor attributes ($p=0$) are handled separately, queries containing only positive or only negative constraints may cause the score to be suppressed by $\alpha$. As Eq. 9 is central to the method, more detailed ablation or sensitivity analysis of $\alpha$ would help clarify the robustness of this design.

2. The method also relies on LLM/VLM models (e.g., Qwen-VL) to extract visual dictionaries and parse user edits. However, the paper does not analyze the accuracy of this step. Variations in attribute coverage and granularity in user queries may affect intent extraction, which could impact retrieval performance and reduce the potential for application in real-world scenarios.

**Strengths And Weaknesses:**

## Strengths
1. The paper proposes a novel structured representation for composed image retrieval by modeling queries and images as visual dictionaries with signed attribute constraints.

2. The framework enables controllable intent modeling and integrates diversity-aware reranking, improving both retrieval accuracy and result diversity.

3. The approach avoids CIR-specific supervision and introduces a self-supervised V-Dict-AE module for learning fine-grained representations.

4. The proposed DFMM-Compose benchmark provides a more comprehensive evaluation of intent satisfaction and diversity.

## Weaknesses
1. **Potential limitation of the scoring design in Eq. 9.** The weights for $p_i$ ($p=+1$) and $n_i$ ($p=-1$) are coupled while $o_i$ ($p=0$) is handled separately, but many realistic queries may only contain positive edits (as in Figure 2) or lack explicit negation, or vice versa. In such cases the score may be dominated by the anchor term $o_i$, potentially weakening the effect of the intended modification. It would be helpful to analyze how the model behaves when queries contain only positive or only negative constraints.
2. **The paper does not provide sufficient analysis of the hyperparameters $\alpha$ and $\beta$ in Eq. 9**. A sensitivity study showing how different values affect retrieval performance would help better understand the model’s behavior and improve reproducibility.
3. **The method relies on Qwen-VL style models to extract structured visual dictionaries and parse user edits into attribute constraints, but the paper does not report the accuracy or robustness of this component**. Since the framework depends on structured key–value representations while real user queries may be ambiguous or unstructured, it would be helpful to either provide quantitative evaluation of the intent extraction accuracy or demonstrate performance under more complex and diverse user queries.

---

> ### Author Rebuttal · Authors · 2026-03-31
>
> We sincerely thank the reviewer for the thoughtful feedback and helpful suggestions. We are encouraged that the reviewer recognizes the novelty and practical value of Pix2Key.
>
> **Q1. Scoring Function Stability**
>
> **A1:** We thank the reviewer for this important question, as it directly reflects the practical motivation behind our scoring design. The open term captures both (i) attributes that remain open in the prompt and (ii) reference-image attributes that are not explicitly marked as desired or undesired. Our intuition is that choosing a reference image already implies weak preference over many unspecified attributes, so these should be treated as **acceptable but non-binding**, rather than irrelevant. Therefore, $o_i$ serves as a soft preservation prior, not a competing edit signal.
>
> To directly address this concern, we split DFMM-Compose into **positive-only**, **negative-only**, and **mixed** query subsets and evaluate them separately. We find that for **positive-only** queries, adding open anchors mainly improves diversity (ILD@50: 52.06 $\rightarrow$ 53.13) while leaving R@50/AC@50 nearly unchanged. For **negative-only** queries, it similarly improves AC@50/ILD@50 with only a small change in R@50 (39.28 $\rightarrow$ 39.06). The clearest gain appears on **mixed** queries, where the full formulation improves all three metrics over removing open anchors (R@50: 39.94 $\rightarrow$ 40.98, AC@50: 53.36 $\rightarrow$ 54.30, ILD@50: 53.21 $\rightarrow$ 54.84). Therefore, the open term does not simply dominate the score; rather, it broadens the acceptable semantic neighborhood and is most helpful when combined with explicit edit-aware constraints.
>
> | Query subset | #Queries | Full setting | R@50 | AC@50 | ILD@50 |
> |---|---:|---|---:|---:|---:|
> | Positive-only | 2008 | pos + open | 41.76 | 54.92 | 53.13 |
> | Negative-only | 303 | neg + open | 39.06 | 53.05 | 52.01 |
> | Mixed | 2194 | pos + neg + open | 40.98 | 54.30 | 54.84 |
>
>
> **Q2. Hyperparameter sensitivity**
>
> **A2:** We thank the reviewer for this important question. In our formulation, however, these parameters are relatively stable: in Eq. (9), $p_i,o_i,n_i$ are all cosine similarities in the same text-embedding space, so $\alpha,\beta$ reweight comparable semantic signals rather than heterogeneous scores; in Eq. (11), relevance and distance are normalized before reranking, which makes $\lambda$ a relatively stable relevance–diversity trade-off knob across queries. Empirically, a small sweep on DFMM-Compose shows only mild variation around the default setting: increasing $\beta$ mainly improves diversity, while changing $\lambda$ affects ILD@50 more than R@50/AC@50. This suggests that the method is not overly brittle and already works well with fixed global settings. Query-adaptive auto-tuning is an interesting extension, but is not required for the current method to generalize effectively.
>
> | Setting | $\alpha$ | $\beta$ | $\lambda$ | R@50 ↑ | AC@50 ↑ | ILD@50 ↑ |
> |---|---:|---:|---:|---:|---:|---:|
> | More edit-focused | 0.60 | 0.20 | 0.20 | 40.88 | 54.21 | 53.47 |
> | Default | 0.50 | 0.30 | 0.20 | **40.96** | **54.44** | 53.96 |
> | More open-preserving | 0.45 | 0.40 | 0.20 | 40.71 | 54.08 | **54.35** |
> | Lower diversity | 0.50 | 0.30 | 0.10 | 40.93 | 54.39 | 52.84 |
> | Higher diversity | 0.50 | 0.30 | 0.30 | 40.81 | 54.22 | **54.88** |
>
> **Q3. VLM-based intent extraction accuracy or robustness**
>
> **A3:** We thank the reviewer for this important question. We provide evidence at both the **task level** and the **dictionary level**. At the task level, the strong **AC@50** results on **DFMM-Compose** already indicate that the retrieved results satisfy the intended edits well, showing that the extracted intent is meaningful and usable in retrieval. In particular, Pix2Key reaches **51.26** AC@50, and **Pix2Key+V-Dict-AE** further improves this to **54.44**, clearly higher than the closest baselines.
>
> At the dictionary level, we also perform a lightweight sanity check on the extracted query dictionaries. Using **GPT-5.4** to judge whether the extracted dictionary matches the original query image and edit prompt, we obtain **94.7%** matching accuracy on all **4,505 DFMM-Compose** samples, further suggesting that the VLM-based intent extraction is reliable in practice.
>
> For robustness under noisier or harder settings, Pix2Key shows consistent stability across several axes. Under vague/noisy instructions (**Reviewer JH17, Q3**), Pix2Key+V-Dict-AE drops by only **0.68**, while strong baselines drop by **2.94–3.78**. Across different query subsets (current Q1) and hyperparameter settings (current Q2), performance also remains stable. Finally, replacing **Qwen2.5-VL-7B** with the much smaller **Qwen2.5-VL-3B** (**Reviewer PeU7, Q3**) causes only a small drop (**R@50: 37.56 $\rightarrow$ 36.94**, **AC@50: 51.26 $\rightarrow$ 50.31**) while substantially improving efficiency. Together, these results suggest that the extracted intent representation is robust in practice.

---

> > ### Author Rebuttal · Reviewer_TU1z · 2026-04-03
> >
> > Thank you for the informative rebuttal.
> >
> > In particular, the clarification in Q1 (Scoring Function Stability) effectively addresses my main concern and improves my understanding of the design.
> >
> > Overall, I consider this work to be technically sound with solid empirical performance. However, I consider the theoretical motivation and conceptual insights somewhat limited, which does not sufficiently motivate me to increase my score. Good luck.

---

> > > ### Author Response · Authors · 2026-04-04
> > >
> > > Dear Reviewer TU1z,
> > >
> > > We sincerely thank you for your thoughtful feedback and for carefully engaging with our work . We truly appreciate your recognition of the technical soundness and empirical strengths of our method. Your questions and suggestions, especially regarding the scoring design and robustness analysis, were very helpful in guiding us to further clarify our motivation and strengthen the presentation. We are grateful for your insightful comments, which helped us better understand the limitations and refine our explanations in the revised version.
> > >
> > >
> > > Best regards,
> > > Authors

---

### Decision · Program_Chairs · 2026-04-30

**Decision:**

Accept (regular)

**Comment:**

This paper addresses Composed Image Retrieval (CIR), where a reference image and text edits guide retrieval. Existing methods fuse image and text into a single embedding, often losing fine-grained attributes and yielding repetitive results. The authors propose Pix2Key, which encodes images and queries as attribute dictionaries and parses edits into constraints for attribute-level matching. Diversity-aware reranking and V-Dict-AE pretraining further enhance expressiveness and result diversity. Unlike triplet-supervised or fused-embedding approaches, Pix2Key’s explicit dictionaries make retrieval more controllable, interpretable, and less dependent on specialized CIR annotations. It presents the DFMM-Compose benchmark to quantitatively evaluate attribute consistency and list diversity. Experiments on multiple benchmarks demonstrate consistent gains in Recall, AC@50, and ILD@50.

A majority of reviewers lean toward positive recommendations. Most of their concerns (e.g., hyperparameter sensitivity, generalization, robustness, efficiency/overhead, and failure modes) have been adequately addressed with additional supporting experiments and analysis. Overall, the paper is technically sound with strengthened empirical validation and is recommended for weak acceptance.